# l-Carnitine Modulates Epileptic Seizures in Pentylenetetrazole-Kindled Rats via Suppression of Apoptosis and Autophagy and Upregulation of Hsp70

**DOI:** 10.3390/brainsci8030045

**Published:** 2018-03-14

**Authors:** Abdelaziz M. Hussein, Mohamed Adel, Mohamed El-Mesery, Khaled M. Abbas, Amr N. Ali, Osama A. Abulseoud

**Affiliations:** 1Department of Medical Physiology, Faculty of Medicine, Mansoura University, Mansoura 35516, Egypt; madel7744@yahoo.com; 2Department of Biochemistry, Faculty of Pharmacy, Mansoura University, Mansoura 35516, Egypt; elmesery@hotmail.com; 3Faculty of Medicine, Mansoura University, Mansoura 35516, Egypt; kmakm_1@outlook.com (K.M.A.); amrnabmans@hotmail.com (A.N.A.); 4Neuroimaging Research Branch, IRP, National Institute on Drug Abuse, National Institutes of Health, Biomedical Research Center, 251 Bayview Blvd, Baltimore, MD 21224, USA; osama.abulseoud@nih.gov

**Keywords:** l-carnitine, PTZ, epilepsy, apoptosis, β-catenin, oxidative stress

## Abstract

l-Carnitine is a unique nutritional supplement for athletes that has been recently studied as a potential treatment for certain neuropsychiatric disorders. However, its efficacy in seizure control has not been investigated. Sprague Dawley rats were randomly assigned to receive either saline (Sal) (negative control) or pentylenetetrazole (PTZ) 40 mg/kg i.p. × 3 times/week × 3 weeks. The PTZ group was further subdivided into two groups, the first received oral l-carnitine (l-Car) (100 mg/kg/day × 4 weeks) (PTZ + l-Car), while the second group received saline (PTZ + Sal). Daily identification and quantification of seizure scores, time to the first seizure and the duration of seizures were performed in each animal. Molecular oxidative markers were examined in the animal brains. l-Car treatment was associated with marked reduction in seizure score (*p* = 0.0002) that was indicated as early as Day 2 of treatment and continued throughout treatment duration. Furthermore, l-Car significantly prolonged the time to the first seizure (*p* < 0.0001) and shortened seizure duration (*p* = 0.028). In addition, l-Car administration for four weeks attenuated PTZ-induced increase in the level of oxidative stress marker malondialdehyde (MDA) (*p* < 0.0001) and reduced the activity of catalase enzyme (*p* = 0.0006) and increased antioxidant GSH activity (*p* < 0.0001). Moreover, l-Car significantly reduced PTZ-induced elevation in protein expression of caspase-3 (*p* < 0.0001) and β-catenin (*p* < 0.0001). Overall, our results suggest a potential therapeutic role of l-Car in seizure control and call for testing these preclinical results in a proof of concept pilot clinical study.

## 1. Introduction

Epilepsy is the second most common neurological issue after stroke affecting around 65 million people around the world [1,2]. As early as around 400 BC, it was proposed in the Hippocratic compositions that seizures start from the mind; however, the connection was not settled until seminal scientific investigations were conducted in the mid-Nineteenth Century by John Hughlings Jackson [3]. From that point forward, epileptology has grown significantly in parallel with fundamental discoveries in neuroscience.

Epilepsy is a neurological disease that is characterized by recurrent epileptic fits [4]. This disease involves several subtypes with particular phenotypes and pathophysiologies that are classified according to their etiology into symptomatic (secondary to other disorders, e.g., cancer or brain insult), idiopathic and cryptogenic (undetermined etiology, probably symptomatic) [5]. Basically, the term ictogenesis means the neurobiological basis for the transition of the brain from the interictal state to the ictal state, whereas epileptogenesis refers to the fundamental processes that lead to the development of the chronic phase of epilepsy with spontaneous recurrent seizures [6]. The molecular mechanisms behind ictogenesis and epileptogenesis are under intense investigation [7]. Current pharmacological treatment strategies largely aim at decreasing neuronal excitability and thereby preventing the occurrence of seizures. An alternative approach is to prevent the emergence of the epileptic state [8]. However, breakthrough seizures, treatment resistance [8], increased epilepsy-related morbidity and mortality [9] are fairly common, highlighting the fact that the efficacy of our current antiepileptogenic medications remains suboptimal [10]. This major public health concern calls for novel therapeutic targets for epilepsy [11]. 

l-carnitine (l-Car) is a dietary supplement that is available in health food stores. It is an endogenous molecule that plays an important role in β-oxidation via transport of acyl-moieties from fatty acids through the mitochondrial membrane [12]. After oral administration of l-Car, its plasma and cerebrospinal fluid (CSF) concentrations increase because it is easily transported through the blood-brain barrier via the organic cation/carnitine transporter OCTN2 [13]. Semland et al. [14] demonstrated that oral administration of l-Car in drinking water failed to stop the development of epilepsy in a kindling animal model, although it was associated with significant improvement of brain metabolism and energy production in pentylenetetrazole (PTZ)-induced epilepsy mouse model. Therefore, the present study was designed to explore the possible antiepileptic role of l-car and its effects on oxidative stress markers, apoptosis, autophagy and the neuroprotective heat shock proteins expression in the PTZ-kindled rat model.

## 2. Materials and Methods

### 2.1. Experimental Animals

Thirty male Sprague Dawely rats, 12–16-week-old at the beginning of the experiment, were housed in individual cages and had free access to food and water with a 12-h light-dark cycle. One week before the experiments, rats were adapted to these conditions with monitoring of the body weight and general conditions throughout the study. All protocols and experimental procedures were approved by the Institutional Review Board (IRB) at Mansoura Faculty of Medicine on 6 February 2017 with Approval Code # r/16.12.90.

### 2.2. Study Groups

Rats were randomly assigned to one of three groups (*n* = 10/group): 1, control negative: received saline (Sal group); 2, control positive pentylenetetrazole (PTZ); (Sal + PTZ group): rats received Sal and PTZ (40 mg/kg i.p. 3-times per week till full kindling) [15]; and 3, l-Car + PTZ group: same as the Sal + PTZ group, but rats received l-Car (100 mg/kg/day via gastric gavage) [16].

### 2.3. Pentylenetetrazole-Kindled Rat Model and Scoring of Epileptic Seizure

For induction of epilepsy, we used the method described by Hansen et al. [15]. After each PTZ injection, each rat was observed for 30 min for latency to epileptic fit, duration of seizure and seizure stage according to a modified Racine scale [15]. Full kindling was defined as exhibiting Stage 4 or 5 of seizure score on three consecutive trials.

### 2.4. Euthanasia and Collection of Brain Samples

Rats achieving full kindling were euthanized the following day after deep anesthesia using sodium thiopental (120 mg/kg i.p.). Five rats in each group were perfused transcardially with 100 mL heparinized saline followed by 150 mL of 10% formalin. The brain was then collected and placed in 10% paraformaldehyde for 4 h for fixation before it was stored in a 25% sucrose plus 0.1% sodium azide solution until processing. For the biochemical assay of markers of oxidative stress and Western blotting, whole brain tissues (of the remaining 5 rats in each group) were collected after saline perfusion only and stored in liquid nitrogen until the time of the required experiments. 

### 2.5. Assay of Lipid Peroxidations Marker (MDA) and Antioxidants (GSH Activity) and Catalase in Brain Tissues

About 50–100 mg of brain tissues were homogenized using a mortar and pestle in 1–2 mL of cold buffer solution (50 mM potassium phosphate, pH 7.5, 1 mM ethylenediaminetetraacetic acid (EDTA)) then centrifuged at 4000 rpm for 15 min at 4 °C. The supernatant was kept at −20 °C until used for analysis. Malondialdehyde (MDA), reduced glutathione (GSH) and catalase enzyme activity in the supernatant of kidney homogenates were measured using a colorimetric method according to the manufacturer’s instructions (Bio-Diagnostics, Dokki, Giza, Egypt). 

### 2.6. Gel Electrophoresis and Western Blotting for Caspase-3 and β-Catenin

Brain tissues were lysed using RIPA buffer solution (50 mM Tris-HCl pH 7.4, 1% triton-X, 0.1% SDS, 150 mM NaCl, 2 mM EDTA, 50 mM NaF) containing protease inhibitor (Roche Diagnostics, Mannheim, Germany) and phosphatase inhibitor mixture II (Sigma Aldrich, Deisenhofen, Germany) to obtain total cell lysates. Afterwards, protein concentrations in each sample were assayed using the Bradford protein assay, and 30 µg of each sample were mixed with Laemmli buffer (pH 8.0) containing 8% sodium dodecyl sulfate (SDS), 0.2 M Tris, 10% β-mercaptoethanol and 40% glycerol. Then, protein samples were denaturated by heating at 95 °C for 5 min. Vertical SDS-PAGE (polyacrylamide) electrophoresis was performed to fractionate total cell lysate proteins according to their molecular weight. Then, the fractionated proteins were transferred to the nitrocellulose membrane by the wet blotting method. Afterwards, membranes were blocked by incubation with 5% BSA solution for 1 h at room temperature. Samples were incubated at 4 °C overnight with primary antibodies specific for caspase-3 (molecular weight (MW) 35 kDa, #9662, Cell Signaling Technology, Danvers, MA, USA), β-catenin (MW 92 kDa, clone 6B3, Cell Signaling Technology, Danvers, MA, USA) and tubulin-α (MW 50 kDa, Neomarkers, Fremont, CA, USA). The secondary antibodies were anti-mouse-HRP (Dako-Cytomation, Glostrup, Denmark) and anti-rabbit-HRP (Cell Signaling Technology, Danvers, MA, USA). Finally, membranes were visualized using the ECL Western blotting detection system (Thermo Fisher Scientific, Waltham, MA, USA) according to the manufacturer’s instructions.

### 2.7. Histopathological Examination of Hippocampal Neurons by Hematoxylin and Eosin

Twenty micrometer-thick sections of brain slices were stained with hematoxylin and eosin (H&E), and the slides of hippocampus were stained with hematoxylin for 15 min and in HCl alcohol solution for 35 s. Then, the sections were immersed with eosin for 10 min and 90% ethanol for 40 s. After that, the section was examined, and images of the cornu ammonis (CA3) region were captured under light microscope.

### 2.8. Measurement of Expression of Heat Shock Protein (Hsp) 70 and Microtubule-Associated Protein 1A/1B-Light Chain 3 (LC3) by Immunohistochemistry in Hippocampus

Serial coronal sections (40 µm) were sliced using a freezing sledge microtome, and a 1:6 series was used for all quantitative immunohistochemistry. Peroxidase-based immunostaining was completed as described previously [17]. In brief, following quenching of endogenous peroxidase activity where appropriate (using a solution of 3% hydrogen peroxide/10% methanol in distilled water) and blocking of nonspecific secondary antibody binding (using 3% normal serum in Tris-buffered saline (TBS) with 0.2% Triton X-100 at room temperature for 1 h), sections were incubated overnight at room temperature with the appropriate primary antibody diluted in 1% normal serum in TBS with 0.2% Triton X-100 (polyclonal anti-LC3 rabbit antibody, Cat#YPA1340, dilution 1:200; Chongqing Biospes Co., Ltd., Chongqing, China) and with primary rabbit polyclonal anti-Hsp70 (Cat #NBP1-35969, Novus Bio, Littleton, CO, USA) (diluted 1:50). Processing of brain sections and the process of immunostaining and image capture and analysis were mentioned in full detail in our previous work [18].

### 2.9. Statistical Analysis

Behavioral data (i.e., seizure scores, time to first seizure and the duration of seizures) are presented as the mean ± standard errors of mean (SEM). Molecular data on the effects of l-Car on oxidants and antioxidants are also presented as the mean ± SEM. Separate repeated measures analyses of variance (ANOVA) with treatment (saline and l-Car) and time (days) factors were used to compare each behavioral variable between control positive (PTZ + saline) and experimental (PTZ + l-Car) groups. When a significant interaction was detected, post-hoc *t*-tests were used to compare the two groups at different time points. Two separate survival analyses were used to examine differences in the time to first seizure and differences in the duration of seizure between the two groups. Separate one-way analyses of variance (ANOVA) were used to compare each molecular variable between control negative (saline), control positive (PTZ + saline) and experimental (PTZ + l-Car) groups. Pearson correlation analyses were used to study the relationships between seizure stage and individual oxidative stress markers in the PTZ group. All data were analyzed using GraphPad Prism Version 7 software, (GraphPad Software, La Jolla, CA, USA). Results were considered significant when *p* ≤ 0.05. 

## 3. Results

### 3.1. The Behavioral Effects of l-Car on PTZ-Induced Seizure

l-Car treatment was associated with marked reduction in seizure score (F (1, 16) = 22.3, *p* = 0.0002) that was evident as early as Day 2 of treatment (l-Car + PTZ vs. Sal + PTZ Day 1 mean ± SEM = 1.2 ± 0.4 vs. 2.4 ± 0.17 *t* = 2.61 df = 16, *p* = 0.018) and continued throughout treatment (Day 14: 1.4 ± 0.5 vs. 4.2 ± 0.2, *t* = 4.64 df = 16, *p* = 0.0003; Figure 1A). Furthermore, l-Car significantly prolonged the time to the first seizure (median survival time l-Car + PTZ vs. Sal + PTZ = 165 vs. 100 s, X^2^ = 31.07, df = 1, *p* < 0.0001; Figure 1B) and shortened seizure duration (median survival time l-Car + PTZ vs. Sal + PTZ = 30 vs. 35 s, X^2^ = 4.81, df = 1, *p* = 0.028; Figure 1C). Furthermore, we studied the relationship between seizure latency and seizure score and found a significant inverse correlation (*r* = −4.96, *p* < 0.0001, *n* = 91; Figure 1D). Moreover, animals with short latency (≤100 s) compared to long latency (>100 s) have a significantly higher seizure score (*t* = 5.739, df = 88, *p* < 0.0001; Figure 1E). 

### 3.2. The Molecular Effects of l-Car on MDA, GSH, Catalase Enzyme, Caspase-3 and β-Catenin

l-Car administration for 14 days attenuated PTZ-induced increase in MDA level (F (2, 15) = 98.51, *p* < 0.0001; Figure 2A), reduced the activity of catalase enzyme (F (2, 15) = 12.76, *p* = 0.0006; Figure 2B) and increased GSH concentration (F (2, 15) = 117.6, *p* < 0.0001; Figure 2C). Moreover, l-Car significantly reduced PTZ-induced elevation in protein expression of caspase-3 (F (2, 15) = 348.6, *p* < 0.0001; Figure 2D) and β-catenin (F (2, 15) = 1813, *p* < 0.0001; Figure 2E). 

### 3.3. Effects of l-Car on the Morphology of Neurons in the CA3 Region of Hippocampus

Brain sections from the Sal group (Figure 3A) showed a normal appearance for neurons in the CA3 region of hippocampus, while those obtained from the Sal + PTZ group (Figure 3B) showed irregular arrangement of neurons with a reduction in the number of neurons, and the neurons showed pyknosis (darkly-stained nucleus and cytoplasm). The number of abnormal neurons was significantly reduced in the brains tissues obtained from the l-Car + PTZ group (Figure 3C).

### 3.4. Effects of l-Car on Hsp70 and LC3 Expression in the CA3 Region of Hippocampus

Immunohistochemical examination showed a significant decrease in the mean area of interest (AOI) of Hsp70-positive cells in the CA3 region of hippocampus in the Sal + PTZ and l-Car + PTZ groups compared to the Sal group (*p* < 0.001). On the other hand, the l-Car + PTZ group showed a significant increase in the number of Hsp70-positive cells compared to the Sal + PTZ group (*p* < 0.01) (Figure 4A). Brain sections showed a high expression of Hsp70 in the Sal group (Figure 4B), mild cytoplasmic expression in the Sal + PTZ group (Figure 4C) and high cytoplasmic expression in the l-Car + PTZ group (Figure 4D).

Furthermore, immunohistochemical examination showed a significant increase in the mean area of interest (AOI) of LC3-positive cells in the CA3 region of hippocampus in the Sal + PTZ group compared to the Sal group (*p* < 0.001). On the other hand, the l-Car + PTZ group showed a significant decrease in the LC3 compared to the Sal + PTZ group (*p* < 0.01) (Figure 5A). Brain sections showed negative expression of LC3 in the Sal group (Figure 5B), high cytoplasmic expression in the Sal + PTZ group (Figure 5C) and minimal cytoplasmic expression in the l-Car + PTZ group (Figure 5D).

### 3.5. Correlations between the Stage of Seizure, Oxidative Stress Markers, Hsp70 Expression and LC3 Expression in Hippocampus

Seizure stage showed significant positive correlations between seizure stage and MDA, LC3, caspase-3 and Hsp70 with significant negative correlations with GSH and CAT (*p* ≤ 0.01). Furthermore, expression of LC3 and Hsp70 showed a significant positive correlation with MDA and a significant negative correlation with CAT (*p* ≤ 0.01). In addition, LC3 expression showed significant positive correlation with Hsp70 expression (*p* < 0.01) (Table 1).

## 4. Discussion

The main findings in the present study are: (a) treatment with pentylenetetrazole (PTZ) caused full kindling of rats, enhanced redox state and upregulated the expression of LC3, Hsp70, caspase-3 and β-catenin in the rat hippocampal CA3 region; (b) chronic daily treatment with l-Car (100 mg/kg) caused significant attenuation in seizure stage and duration and reversed the PTZ-induced alterations in the redox sate and the expression of caspase-3, β-catenin and more upregulation of Hsp 70 expression in the rat hippocampal CA3 region.

Acetyl l-Car is a well-known potent antioxidant and has several properties that may suggest its antiepileptic effect, such as protection against glutamate toxicity [19], a favorable effect on energy homeostasis [20,21,22] and lowering oxidative damage and improving mitochondrial function [23]. Moreover, pretreatment with l-Car before administration of a single convulsive dose of PTZ to mice prolonged latency to seizures and reduced the frequency of tonic-clonic seizures in a dose-dependent manner [24]. On the other hand, Smeland et al. [14] reported in a model of chronic epilepsy (PTZ-kindled epileptic rat) that l-Car failed to slow down the progression and latency to clonic convulsions. However, they reported that it normalized some metabolic parameters in brain tissues such as lactate, (3–13C) alanine, dopamine, *myo*-inositol and succinate in the cortex.

In the present study, the kindling model (a chronic widely-accepted animal model of epilepsy produced by chemical or electrical stimuli) is implicated; of the chemical stimuli, pentylenetetrazole (PTZ) (a gamma amino butyric acid (GABA) A receptor antagonist) produces severe convulsions when administered to animals by using subconvulsive doses that are applied intermittently and repetitively to produce full-blown convulsions [25]. The present study reported a significant increase in seizure score and duration with significant reduction in latency in the PTZ group. These findings are in line with our previous work from our group [18] and others [26,27]. Furthermore, the present study demonstrated that treatment with l-Car in the PTZ-kindling model greatly attenuated tonic-clonic convulsions and the duration of these convulsions, as well as delayed the seizure onset. 

It has been demonstrated that reactive oxygen species play a role in the development and progression of epileptic seizure [18,28]. During convulsive seizure, the low antioxidant defense systems can predispose the brain to oxidative stress. Furthermore, the hippocampus may be particularly sensitive to oxidative stress because it has low endogenous levels of vitamin E, an important biochemical antioxidant, relatively to other brain regions [29]. The present study demonstrated a high oxidative stress state in the CA3 region of the hippocampus of PTZ kindled rats in the form a significant increase in MDA (marker of lipid peroxidation) and a significant reduction in antioxidants such as catalase and reduced glutathione (GSH). These findings are in line with previous studies [18,27,30] and suggest that the induction of seizure produces a state of oxidative stress. Furthermore, our results provide good evidence for the role of ROS in the pathophysiology of seizure in this model.

In the present study, co-treatment with l-Car significantly reduced MDA level and increased GSH and catalase levels relative to the PTZ group. In agreement with these findings, previous studies reported that l-car attenuated the seizure duration and severity in pilocarpine-induced epilepsy though its antioxidant effects, which involved the suppression of MDA and upregulation of CAT and SOD activities [31]. These findings lend more evidence to the hypothesis that the antiepileptic effect of l-Car could be partially related to its antioxidant effects.

Autophagy is a regulated process in which the intracellular cellular proteins aggregate and damaged organelles are degraded by the cell lysosomes [32]. LC3 is considered the most reliable cellular marker for autophagy activation [33]. Therefore, we evaluated in the present study LC3 protein expression in the CA3 region of hippocampus by immunohistochemistry. According to our results, little expression of LC3 in the CA3 region in the saline negative control group was demonstrated, while the PTZ group showed marked LC3 expression in brains. Furthermore, there was a positive correlation between LC3 expression and seizure stage, suggesting a role for autophagy in PTZ-induced epilepsy. In agreement with these findings, Shacka et al. [34] showed significant accumulation of LC3-positive autophagy vacuoles in the hippocampus of kainite-induced epileptic mice. Furthermore, Cao et al. [35] demonstrated the presence of autophagy in pilocarpine-induced status epilepticus (SE) models. In addition, Scherz-Shouval and Elazar [36] and Scherz-Shouval et al. [37] hypothesized that reactive oxygen species are the most important activators of autophagy. In the present study, we reported a positive correlation between autophagy and MDA level, which is in line with the previous hypothesis. Moreover, in the present study, we demonstrated for the first time that l-Car inhibits autophagy in PTZ-induced chronic epilepsy.

The role of apoptosis in epilepsy was demonstrated in many previous studies and in different animal models. Sudha et al. [28] and Simonian et al. [38] demonstrated activation of apoptosis in hippocampal regions of KA-induced epilepsy. Moreover, Naseer et al. [39] showed a significant increase in caspase-3 expression with activation of neuronal apoptosis in PTZ-induced epilepsy in adult rats. In line with these previous studies, we demonstrated a significant increase in caspase-3 expression in the CA3 region of hippocampus of the PTZ group when compared to the saline control group. Moreover, treatment with l-Car decreased apoptosis significantly in the hippocampus CA3 region compared to the PTZ group. Activation of apoptosis in PTZ-induced epilepsy could be due to oxidative stress and generation of reactive oxygen species, which disrupt mitochondrial membrane potential and activate the mitochondrial pathway for apoptosis [40]. These findings, along with others suggest a potential role for apoptosis in epilepsy and possible anti-apoptotic effects for l-Car in the epileptic rat model.

Heat shock proteins (HSPs) are stress-induced proteins that play an important role in cellular responses to stress [41]. The expression of HSPs has been detected in different types of cells in the nervous system including neuralgia, neurons and endothelial cells [42]. Previous studies examined the expression of Hsp70 in epilepsy and reported direct relationship between seizure frequency, duration, intensity and Hsp70 expression in both animal models [18,43,44] and human epilepsy [45,46]. It was suggested that the expression of Hsp70 has a protective role for brain cells during epilepsy. The present study demonstrated significant upregulation in Hsp70 expression in the hippocampal CA3 region with positive correlations with the seizure stage. Moreover, we demonstrated that l-Car treatment caused upregulation in Hsp70 in hippocampus, which was reflected in improvement of behavioral changes. These results suggest that upregulation of Hsp70 might be one of the potential mechanism of the antiepileptic effects of l-Car. It is plausible to assume that the highly expressed Hsp70 could protect neurons against oxidative stress and apoptosis. In accordance with this assumption, Li et al. [47] and Zhao et al. [48] reported that the highly inducible Hsp70 has a neuroprotective role in hindering apoptosis, possibly through interacting with p53, which elicits the apoptotic process [49,50]. Furthermore, Kanitkara and Bhonde [51] reported that Hsp-70 can reduce oxidative stress in beta cells and increase glucose-induced insulin release. Moreover, Ayala and Tapia demonstrated that induction of Hsp70 protects the hippocampal neurodegeneration via modulating endogenous glutamate expression [52]. A previous study has shown increased hippocampal CA1 glutamate release in PTZ-treated animals that was inhibited by Chai-Long-Ku-Li-Tan, a Chinese herbal medicine, as a mechanism of its anticonvulsant properties [53]. Examining the effect of l-Car on glutamatergic neurotransmission is highly indicated, however beyond the scope of the current study, and requires further investigation.

In the development of the nervous system, the Wnt-β-catenin signaling pathway plays an important role in neurogenesis, neural differentiation, synapse development and plasticity [54,55]. The Wnt signaling pathway might be involved in a number of CNS disorders such as Alzheimer’s disease, schizophrenia and mood disorders [56,57,58]. Regarding epilepsy, the role of the Wnt-β-catenin signaling pathway showed controversies. Some studies showed upregulation in β-catenin in epileptic animal models such as pilocarpine-induced status epilepticus [59], electroconvulsive seizures rat [60] and Theihaber et al. [61], who reported upregulation of β-catenin in the hypoxic rat model. In addition, Xing et al. [62] demonstrated that β-catenin may not be involved in the development of hippocampal sclerosis of mesial temporal lobe epilepsy. On the other hand, other studies demonstrated downregulation of β-catenin during epileptic animal models such as the Kainite-induced rat model [63,64]. Moreover, Campos et al. [65] demonstrated that β-catenin knockout mice have high seizure susceptibility to PTZ. In the present study, we demonstrated upregulation of β-catenin in CA3 hippocampal regions in the PTZ group, which could be in response to oxidative stress. Downregulation of β-catenin in the l-Car group compared to the PTZ group supports this hypothesis. 

## 5. Conclusions

The results of the present study confirmed the neuroprotective and antiepileptic action of l-Car against PTZ-induced epilepsy. These effects might be explained on the basis of detected l-Car actions such as antioxidant activity, anti-apoptotic effects, suppression of autophagy and upregulation of neuroprotective heat shock protein. However, further knockout experiments will be required to study the detailed molecular mechanism of l-Car. 

## Figures and Tables

**Figure 1 brainsci-08-00045-f001:**
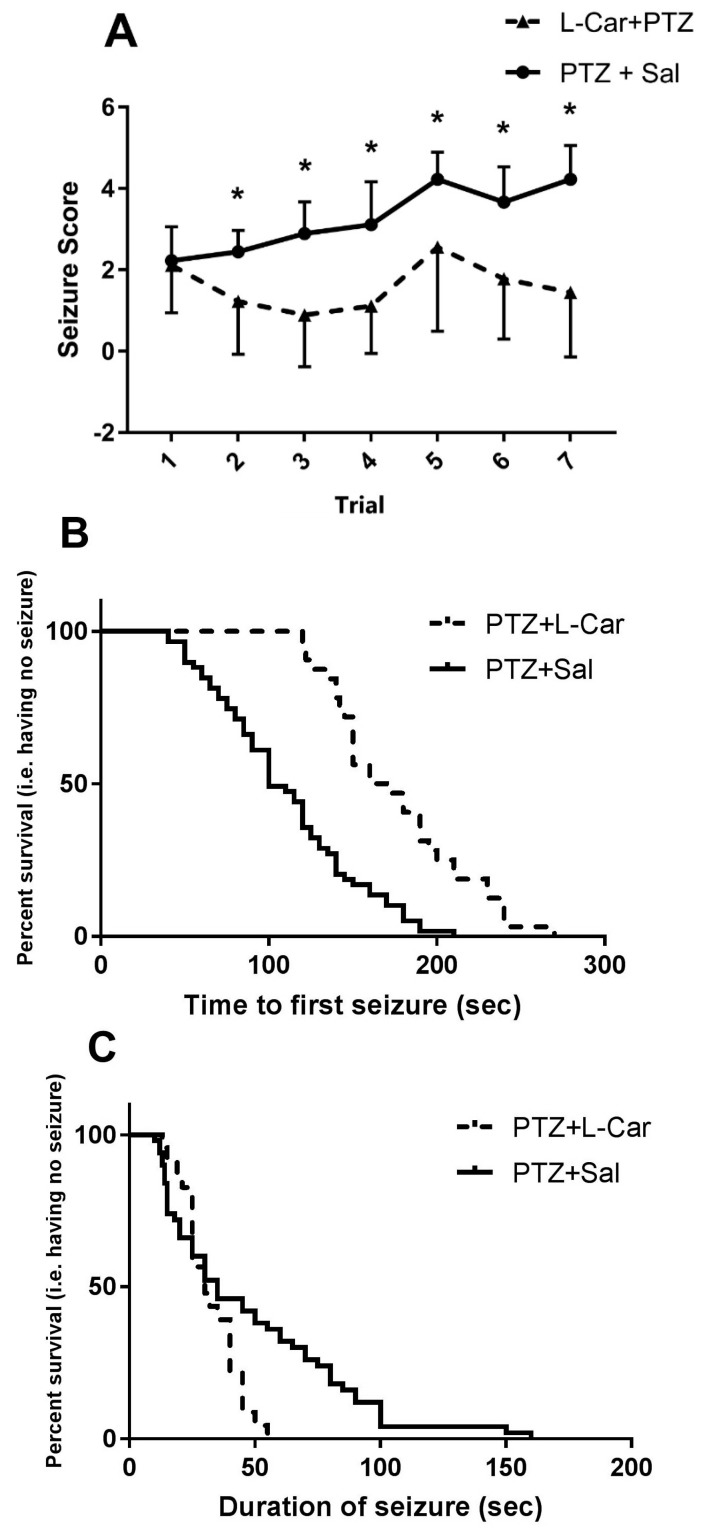
The behavioral effects of l-Car on PTZ-induced seizures. (**A**) Two-way ANOVA showed significant effects of treatment (F_1, 16_ = 22.35, *p* = 0.0002), time (F_6, 96_ = 6.07, *p* < 0.0001) and the interaction between the two factors (F_6, 96_ = 3.15, *p* = 0.007) with significant differences (*) in seizure scores between l-Car + PTZ and Sal + PTZ groups on Day 2 (1.22 ± 0.43 vs. 2.4 ± 0.17, df = 16, *p* = 0.018), Day 3 (0.88 ± 0.42 vs. 2.8 ± 0.26, df = 16, *p* = 0.001), Day 4 (1.1 ± 0.38 vs. 3.1 ± 0.35, df = 16, *p* = 0.005), Day 5 (2.56 ± 0.6 vs. 4.2 ± 0.2, df = 16, *p* = 0.035), Day 6 (1.7 ± 0.49 vs. 3.6 ± 0.28, df = 16, *p* = 0.004) and Day 7 (1.4 ± 0.5 vs. 4.2 ± 0.2, *t* = 4.64, df = 16, *p* = 0.0003) by the two-tailed *t*-test, nine animals per group. (**B**) Survival analysis shows significant delay in time to first seizure in l-Car + PTZ vs. Sal + PTZ animals. Median survival ratio = 1.65 (165 vs. 100 s), 95% CI of ratio = 1.073–2.537, *p* < 0.0001. (**C**) Seizure duration is significantly reduced in l-Car + PTZ vs. Sal + PTZ animals. Median survival ratio is 0.85% (30 vs. 35 s), 95% CI of ratio = 0.523–0.404, *p* = 0.028. (**D**) Correlation between seizure latency and seizure score (*r* = −4.96, *p* < 0.0001, *n* = 91). (**E**) Seizure score in animals with short latency (≤100 s) and long latency (>100 s) (*t* = 5.739, df = 88, *p* < 0.0001). (*) significant in seizure scores between rats with sort latency and long latency. PTZ: pentylenetetrazole; Sal: saline; l-Car: l-carnitine.

**Figure 2 brainsci-08-00045-f002:**
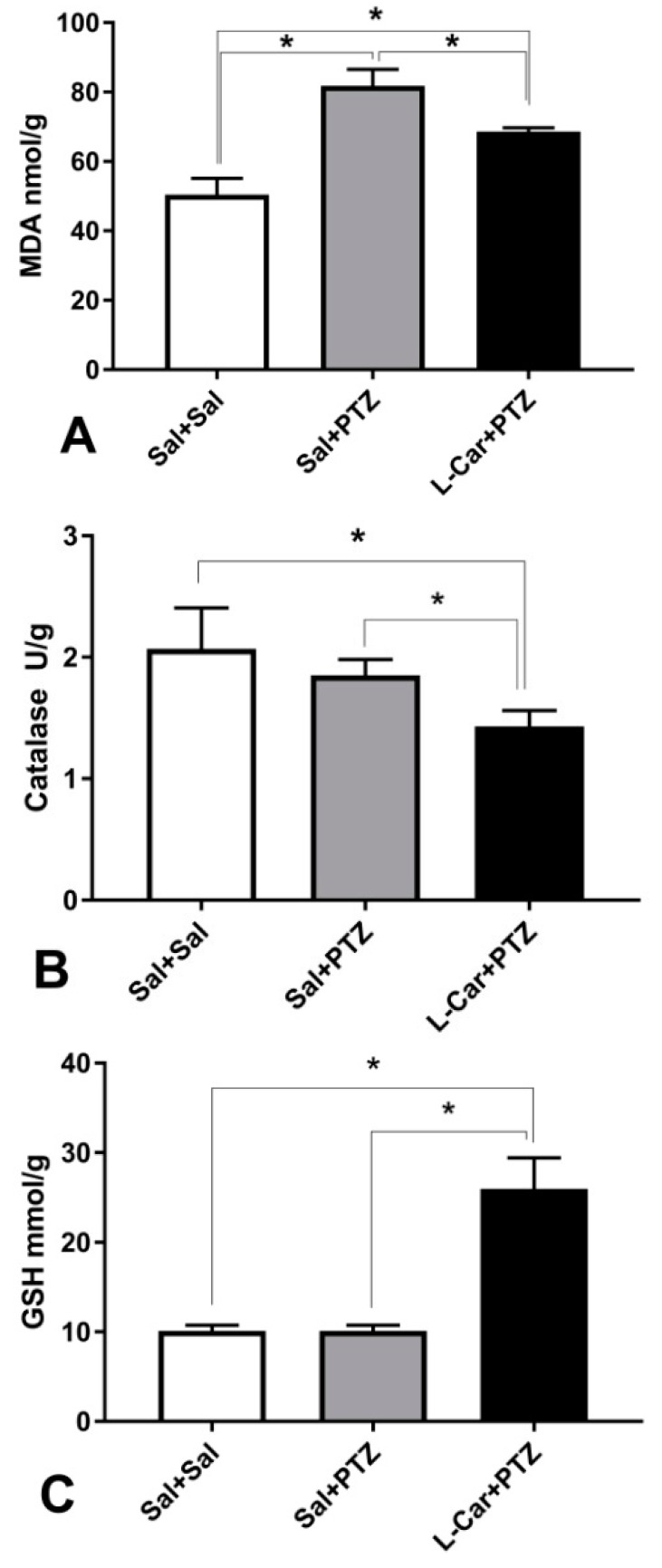
The molecular effects of l-car on oxidants and antioxidants. (**A**) PTZ-treated animals had significantly (*) higher malondialdehyde (MDA) compared to saline controls (*): mean difference = 31.35, 95% CI = 25.52–37.18, *p* < 0.0001, and l-Car-treated PTZ animals compared to saline controls (*): mean difference = 18.22, 95% CI = 12.39–24.04, *p* < 0.0001, as well as compared to PTZ-treated animals: mean difference = −13.13, 95% CI = −18.96–−7.307, *p* < 0.0001, by one-way ANOVA followed by Tukey’s multiple comparisons test, six animals per group. (**B**) Significant difference in catalase enzyme activity (F_2, 15_ = 12.76, *p* = 0.0006) by one-way ANOVA with marked reduction in l-Car-treated animals (compared to saline control (*): mean difference = −0.63, 95% CI = −0.97–−0.30, *p* = 0.0005, and compared to PTZ-treated animals (*): mean difference = −0.42, 95% CI = −0.75–−0.08, *p* = 0.013). No significant difference between PTZ-treated animals and saline controls (mean difference = −0.21, 95% CI = −0.55–0.11, *p* = ns) by Tukey’s multiple comparisons test, six animals per group. (**C**) Robust increase in GSH concentration (F_2, 15_ = 117.6, *p* < 0.0001) by one-way ANOVA. l-Car-treated animals showed a marked increase compared to saline-treated animals (*): mean difference = 15.8, 95% CI = 12.7–18.9, *p* < 0.0001, and compared to PTZ-treated animals (*): mean difference = 15.8, 95% CI = 12.7–18.9, *p* < 0.0001. No significant difference between PTZ-treated animals and saline controls (mean difference = 0, 95% CI = −0.31–3.1, *p* = ns) by Tukey’s multiple comparisons test, six animals per group. (**D**) Significant difference between groups in caspase-3 protein expression (F_2, 15_ = 348.6, *p* < 0.0001) by one-way ANOVA. PTZ increased the expression compared to saline controls (*): mean difference = 0.72, 95% CI = 0.65–0.80, *p* < 0.0001. Furthermore, l-Car increased the expression of caspase-3 protein compared to saline controls (*): mean difference = 0.60, 95% CI = 0.53–0.68, *p* < 0.0001. However, l-Car treatment attenuated the PTZ-induced increase in caspase-3 protein expression (*): l-Car + PTZ vs. Sal + PTZ mean difference = −0.12, 95% CI = −0.197–−0.043, *p* = 0.002, by Tukey’s multiple comparisons test, six animals per group. (**E**) Marked effect of treatment on β-catenin (F_2, 15_ = 1813, *p* < 0.0001) by one-way ANOVA. PTZ increased the expression of β-catenin compared to saline control (*): mean difference = 1.28, 95% CI = 1.23–1.34, *p* < 0.0001. Furthermore, l-Car increased the protein expression of β-catenin compared to saline controls (*): mean difference = 0.48, 95% CI = 0.43–0.54, *p* < 0.0001. However, l-Car treatment attenuated the PTZ-induced increase in β-catenin protein expression (*): l-Car + PTZ vs. Sal + PTZ mean difference = −0.80, 95% CI = −0.85–−0.74, *p* < 0.0001, by Tukey’s multiple comparisons test, six animals per group, (**F**) Products of Western blot for caspase-3, β-catenin protein and tubulin (housekeeping gene) protein expression in the Sal group [1], Sal + PTZ group [2] and l-Car + PTZ group [3].

**Figure 3 brainsci-08-00045-f003:**
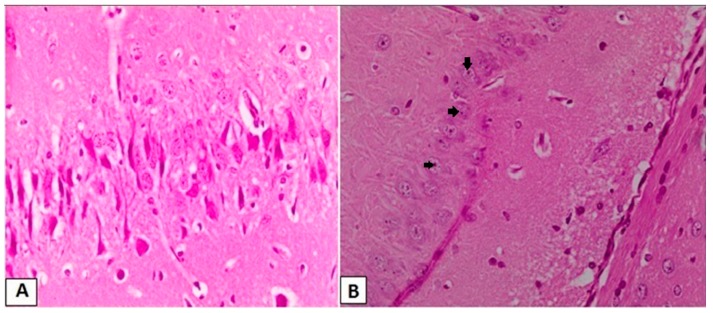
Histopathological examination of the CA3 region in hippocampus. Normal morphology for cells and adequate number in CA3 in brains obtained from the Sal group (**A**, H&E, 400×), reduced number of neurons with pyknotic changes (black arrows) in neurons of CA3 in brains obtained from the Sal + PTZ group (**B**, H&E, 400×) and significant number of neurons with normal morphology in CA3 in brains obtained from the l-Car + PTZ group (**C**, H&E, 400×).

**Figure 4 brainsci-08-00045-f004:**
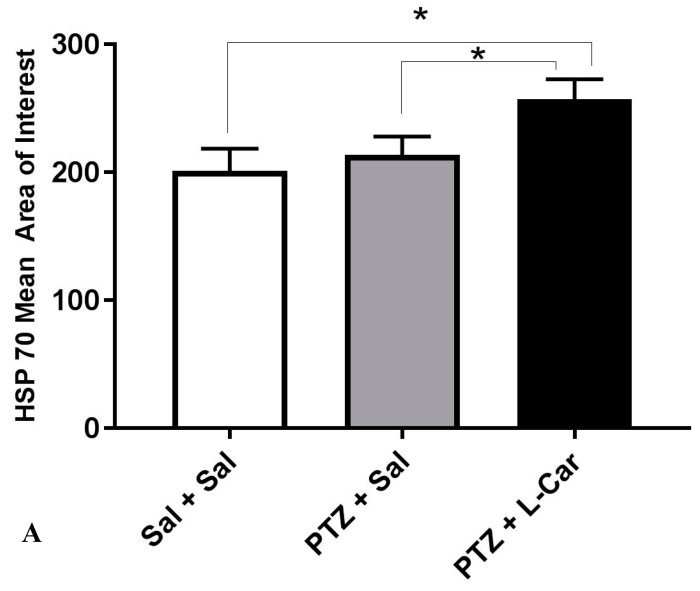
Mean area of interest of Hsp70 positivity in the CA3 region of hippocampus from different groups (**A**). The HSP70 protein expression mean area of interest was compared between the three groups (PTZ + Sal, Sal + Sal and PTZ + l-Car using one-way ANOVA. F(2,15)=20.53, *p* < 0.0001. Tukey multiple comparisons test was done between the groups. There was a significant difference between PTZ + Sal vs. PTZ + l-Car (*) mean difference = −43.67. 95% CI of difference = −67.47 to −19.86, *p* = 0.0007) and also between Sal + Sal vs. PTZ + l-Car (*) mean difference = −55.83. 95% CI of difference = −79.64 to −32.03, *p* < 0.0001. Section of brain in the CA3 region of hippocampus showing negative expression for Hsp70 in the Sal group (**B**, 200×), moderate cytoplasmic expression in the Sal + PTZ group (**C**, 200×) and high cytoplasmic brown staining for Hsp70 in the l-Car + PTZ group (**D**, 200×).

**Figure 5 brainsci-08-00045-f005:**
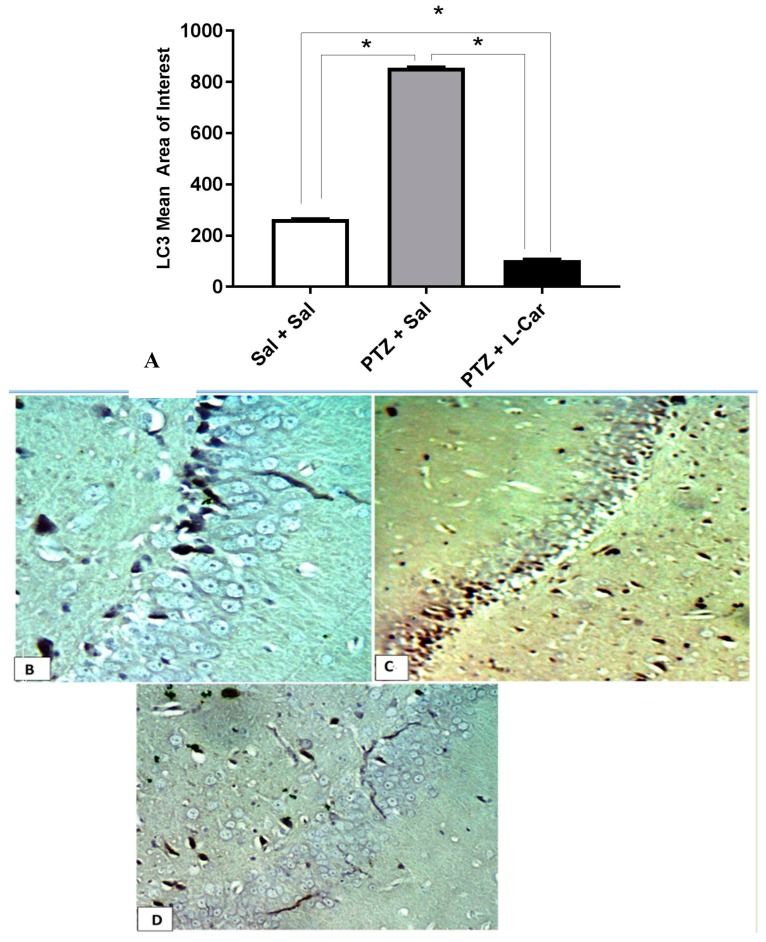
Mean area of interest of LC3 positivity in the CA3 region of hippocampus from different groups (**A**). LC mean area of interest by one-way ANOVA F(2,15) = 72,703, *p* < 0.0001. Tukey multiple comparisons test was done between the groups. There was a significant difference between PTZ + Sal vs. PTZ + l-Car (*) mean difference = 750.8. 95% CI of difference = 745.5 to 756.2, *p* < 0.0001 and also between Sal + Sal vs. PTZ + l-Car (*) mean difference = 160.4. 95% CI of difference = 155 to 165.8, *p* < 0.0001, and also between PTZ + Sal vs. Sal + Sal (mean difference = 590.4 95% CI of difference = 585 to 595.8, *p* < 0.0001). Section of brain in the CA3 region of hippocampus showing negative expression for LC3 in the Sal group (**B**, 400×), high cytoplasmic expression in the Sal + PTZ group (**C**, 400×) and low expression for LC3 in the l-Car + PTZ group (**D**, 400×).

**Table 1 brainsci-08-00045-t001:** Correlation between seizure stage and oxidative stress markers, expression of caspase-3, β-catenin, LC3 and Hsp70 in the PTZ group.

Parameters		Stage of Seizure	MDA	GSH	CAT	Caspase-3	β-Catenin	Hsp70	LC3
Seizure stage	*r*		0.78	−0.74	−0.34	0.65	0.63	−0.61	0.67
	*p*		0.001	0.02	0.092	0.005	0.03	0.04	0.011
MDA	*r*			−0.811	−0.44	0.88	0.67	−0.74	0.75
	*p*			0.01	0.067	0.005	0.01	0.01	0.01
GSH	*r*				0.32	−0.82	0.85	0.67	−0.71
	*p*				0.21	0.005	0.001	0.004	0.003
CAT	*r*					−0.13	−0.32	0.64	−0.67
	*p*					0.62	0.21	0.01	0.009
Caspase-3	*r*						0.68	−0.62	0.68
	*p*						0.01	0.04	0.03
β-catenin	*r*							0.63	0.62
	*p*							0.01	0.01
Hsp70	*r*								0.62
	*p*								0.011

MDA = malondialdehyde, GSH = reduced glutathione concentration, CAT = catalase activity, Hsp70 = heat shock protein 70. *r*, Pearson’s correlation coefficient. *p* < 0.05 is considered significant.

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
