# Peer review of "l-Carnitine Modulates Epileptic Seizures in Pentylenetetrazole-Kindled Rats via Suppression of Apoptosis and Autophagy and Upregulation of Hsp70"

_brainsci, 2018, doi:10.3390/brainsci8030045_

Round 1

Reviewer 1 Report

BS-272930

The authors demonstrated that L-carnitine protected against PTZ-induced seizures and neurotoxicity via antioxidant activity, suppression of apoptosis and autophagy, and upregulation of Hsp70. The results are interesting, and may provide readers with good information.

There are no data of neurotransmitters which may govern seizures. So, it is not clear that neuroprotective parameters such as MDA GSH, caspase, and so on are causative factors or results of epilepsy control. It is recommended that additional analysis of glutamate and/or GABA and more careful interpretation are required.

Please describe manufacturers and their country for reagents and devices. For example, --- (Sigma-Aldrich, St. Louis, MO, USA) ---.

Full names should be described for the first time, followed by abbreviations in parenthesis. For example, pentylenetetrazole (PTZ).

The quality of microphotographs should be improved. Especially, please check the photos A, B, and C if they were not exchanged. Actually, the dark pyknotic cells are dead.

Please check the format of reference list. It should be described according to the Instructions for authors.

Author Response

Reviewer # 1

The authors demonstrated that L-carnitine protected against PTZ-induced seizures and neurotoxicity via antioxidant activity, suppression of apoptosis and autophagy, and upregulation of Hsp70. The results are interesting, and may provide readers with good information.

Comment

There are no data of neurotransmitters which may govern seizures. So, it is not clear that neuroprotective parameters such as MDA GSH, caspase, and so on are causative factors or results of epilepsy control. It is recommended that additional analysis of glutamate and/or GABA and more careful interpretation are required.

Response

     We thank the reviewer for raising this point. A previous study by Wu et al., 2000 (The Chinese herbal medicine Chai-Hu-Long-Ku-Mu-Li-Tan (TW-001) exerts anticonvulsant effects against different experimental models of seizure in rats. Wu HM1, Huang CC, Li LH, Tsai JJ, Hsu KS, Jn J pharmacol 2000) has shown increased hippocampal CA1 glutamate release in PTZ-treated animals that was inhibited by Chai-Long-Ku-Li-Tan, a Chinese herbal medicine as a mechanism of its anticonvulsant properties. However, in this current study we aimed to examine the efficacy of L-Carnitine in attenuating PTZ-induced seizures and probe the molecular mechanisms through measuring antioxidant markers.  We added this limitation to the discussion section and we will follow with another study focusing primarily on the glutamatergic effects of L-Carnitine.

Comment

Please describe manufacturers and their country for reagents and devices. For example, --- (Sigma-Aldrich, St. Louis, MO, USA) ---.

Response

We apologize for forgetting to mention the details data of manufacturers. Please, check the modification in red color in the manuscript.

Comment

Full names should be described for the first time, followed by abbreviations in parenthesis. For example, pentylenetetrazole (PTZ).

Response

The manuscript was edited to ensure that we mention the full name first then the abbreviation

Comment

The quality of microphotographs should be improved. Especially, please check the photos A, B, and C if they were not exchanged. Actually, the dark pyknotic cells are dead.

Response

The resolution of micrographs of the pathology was improved and enahced in revised manuscript and black arrows were added to show pyknotic cells which are dead cells in slices of PTZ+ Sal group.

Comments

Please check the format of reference list. It should be described according to the Instructions for authors.

Response

All references we corrected according of the journal reference style

Reviewer 2 Report

In this study, using PTZ-kindled model of epilepsy, the authors investigated effect  of  L-carnitine on seizures, on oxidative stress, apoptosis, autophagy and heat shock protein (HSP) expression. They showed L-Car treatment at a dose of 100mg/Kg daily caused a reduction in seizure severity and duration, redox state, and expression of aspase-3, beta catenin, as well as an increase in HSP70 in the CA3 region of hippocampus. They concluded that L-car has possible antiepileptic effect on PTZ-induced epilepsy model and that this action maybe mediated through its antioxidant activity, anti-apoptotic effects, suppression of autophagy and upregulation of neuroprotective heat shock protein.

Here are my comments.

1. It would be interesting to know if L-Car treatment  affected epileptogenesis . Does the  treatment affect the latency for the PTZ-kindled rats to reach full kindling? If the rats have chronic spontaneous epilepsy after kindling, are there any differences in seizure severity, duration and frequency between L- Car and saline- treated animals?

2. Please clarify the age of the rats and sample size for each experiment.

3. More details including references are needed to describe the method used for the quantification of cell number, HSP70  and LC3 in AOI of CA3( Fig 3, 4 & 5).

4. Figure 2 only shows representative immuno images. To make it more convincing, quantification needs to be done.

5. Please clarify survival analysis used for Fig 1B, C. Units are needed for X axis.

6. Please clarify “$” used in legend of fig 4& 5.

7. Line254-259 does not match the corresponding figure 5.

8. Line 280-283 needs to be edited, “treatment with PTZ caused significant increase in Racine score……..decrease in latency of onset”. What is the comparison mentioned here? PTZ vs saline group that is not supposed to have seizures?

9. Please have a native English speaker thoroughly edit the manuscript prior to re-submission because there are many typos grammatic errors.

Author Response

Author reply

We thank the editor and the reviewers for their valuable comments that improved the quality of our manuscript significantly. Here we provide point-by-point response to their questions. We have also edited the language of the manuscript as requested. 

Reviewer # 2

n this study, using PTZ-kindled model of epilepsy, the authors investigated effect  of  L-carnitine on seizures, on oxidative stress, apoptosis, autophagy and heat shock protein (HSP) expression. They showed L-Car treatment at a dose of 100mg/Kg daily caused a reduction in seizure severity and duration, redox state, and expression of aspase-3, beta catenin, as well as an increase in HSP70 in the CA3 region of hippocampus. They concluded that L-car has possible antiepileptic effect on PTZ-induced epilepsy model and that this action maybe mediated through its antioxidant activity, anti-apoptotic effects, suppression of autophagy and upregulation of neuroprotective heat shock protein.

Here are my comments.

Comments

1. It would be interesting to know if L-Car treatment affected epileptogenesis. Does the treatment affect the latency for the PTZ-kindled rats to reach full kindling? If the rats have chronic spontaneous epilepsy after kindling, are there any differences in seizure severity, duration and frequency between L- Car and saline- treated animals?

Response

We thank the reviewer for his question and we have shown under the results section that indeed L-Car treatment was associated with marked reduction in seizure severity as measured by seizure score [F (1, 16) = 22.3, P=0.0002] that was evident as early as day 2 of treatment [L-Car+PTZ vs. Sal+PTZ day 1 mean ± SEM = 1.2 ± 0.4 vs. 2.4 ± 0.17 t=2.61 df=16, P=0.018] and continued throughout treatment [day 14: 1.4 ± 0.5 vs. 4.2 ± 0.2, t=4.64 df=16, P = 0.0003, Fig 1A]. Furthermore, L-Car significantly prolonged the time to the first seizure [median survival time L-Car+PTZ vs. Sal+PTZ=165 vs. 100 seconds, X2 = 31.07, df = 1, P < 0.0001, Fig 1B] and shortened seizure duration [median survival time L-Car+PTZ vs. Sal+PTZ=30 vs. 35 seconds, X2 = 4.81, df = 1, P=0.028, Fig 1C].

Comment

2. Please clarify the age of the rats and sample size for each experiment.

Response

The age and number of rats for each experiment were added to revised manuscript

Comment

3. More details including references are needed to describe the method used for the quantification of cell number, HSP70  and LC3 in AOI of CA3( Fig 3, 4 & 5).

Response

Reference was added to methods section in revised manuscript

Comment

4. Figure 2 only shows representative immuno images. To make it more convincing, quantification needs to be done.

Response

Figure 2 represents Western Blotting anaylsis of caspase-3 and β-catenin. Regarding β-catenin, we already did quantification for its expression as indicated in the figure. Regarding caspase-3, there was no indicated differences in its expression in the different groups and we did not detect any splitted products in the different groups. Therefore, there was no need to convert Western Blotting results of caspase-3 in a quantitative figure as in case of β-catenin.

Comment

5. Please clarify survival analysis used for Fig 1B, C. Units are needed for X axis.

Response

We have clarified the X-axis units in the survival analysis curves as requested

Comment

6. Please clarify “$” used in legend of fig 4& 5.

Response

We thank you for this comment. This is a typing error which is corrected in revised manuscript

Comment

7. Line254-259 does not match the corresponding figure 5.

Response

This is corrected

Comment

8. Line 280-283 needs to be edited, “treatment with PTZ caused significant increase in Racine score……..decrease in latency of onset”. What is the comparison mentioned here? PTZ vs saline group that is not supposed to have seizures?

Response

This sentence was edited in revised manuscript

Comment

9. Please have a native English speaker thoroughly edit the manuscript prior to re-submission because there are many typos grammatic errors.

Response

Done

Round 2

Reviewer 2 Report

The revised manuscript has got improved significantly.

Please read my previous comment 1 carefully and then respond. Please be noted the comment was about the effect on latency for the rats to reach full kindling and the effect on chronic spontaneous seizures.

Also, in figure 3, there are only representative immuno images. Quantificaiton needs to be done.

Author Response

Authors Reply

We thank the editor and the reviewers for their valuable comments that improved the quality of our manuscript significantly. Here we provide point-by-point response to their questions. We have also edited the language of the manuscript as requested. 

Comment 1

Please read my previous comment 1 carefully and then respond. Please be noted the comment was about the effect on latency for the rats to reach full kindling and the effect on chronic spontaneous seizures.

Response

Thank you. We added new correlation between seizure and latency and found significant inverse correlation [r=-4.96, P<0.0001, n=91, Fig.1D]. Moreover, we found that animals with short latency (≤100 sec) compared to long latency (>100 sec) have significantly higher seizure score [t=5.739, df=88, P<0.0001, Fig 1E). These figures suggest that the shorter the latency the higher the degree of seizure score. Regarding the effect of L-car on spontaneous seizure we did not record the spontaneous seizure because this requires 24 hrs recording for rats which is not easy to do in our lab. However, we found in the present that L-CAR treatment with associated with reduction in seizure score and prolongation of seizure latency.

Comment 2

Also, in figure 3, there are only representative immuno images. Quantification needs to be done.

Response

Thanks again for your comments to improve the quality of our paper. Figure 3 is a figure of brain specimen stained with hematoxylin and Eosin. The aim of this figure is to observe morphological alterations of the hippocampal neurons and to demonstrate the significant change in number of neurons in rats treated with PTZ not quantification of the expression of some marker like immunostaining in figures 4 and 5.